# Nudging the Boundaries of LLM Reasoning

**Justin Chih-Yao Chen**[1,2]    **Becky Xiangyu Peng**[1]    **Prafulla Kumar Choubey**[1]

**Kung-Hsiang Huang**[1]    **Jiaxin Zhang**[1]    **Mohit Bansal**[2]    **Chien-Sheng Wu**[1]

[1]Salesforce AI Research    [2]UNC Chapel Hill

## Abstract

Current online reinforcement learning (RL) algorithms like GRPO share a key limitation in LLM reasoning: they cannot learn from problems that are "unsolvable" to the model. In other words, they can only improve performance on problems where the model is capable of exploring the correct answer. If a problem is too difficult–such that even hundreds of attempts never produce a correct solution–the model cannot learn from it. Consequently, the model's "upper limit" remains unchanged after RL training, even though the likelihood of solving easier, solvable problems may increase. These hard, unsolvable samples–though potentially rich in learning signal–cannot contribute to training, as no rollouts yield rewards and thus no gradients are produced. To unlock learning from these hard samples, we propose NuRL [1], a "nudging" method that aims to push the upper bound of LLM reasoning using self-generated hints, i.e., abstract cues that help reduce the problem difficulty for the model. Given a question and its gold answer, the model generates a Chain-of-Thought (CoT) and then produces a hint containing the core knowledge needed to solve the problem. During online RL training, we generate $\mathcal{G}$ rollouts from the base policy and use the pass rate to decide whether the hint should be injected. For hard samples with a 0% pass rate, we inject the offline-generated hint and regenerate a new batch of trajectories. This yields two benefits: (1) the hint boosts pass rates (from 0% to non-zero), thereby introducing training signals for previously unsolvable samples, and (2) the hints are self-generated (conditioned on the gold answer), avoiding distributional shift and do not rely on external models. Compared to standard GRPO, NuRL achieves consistent improvements across six diverse benchmarks and three models, while remaining complementary to test-time scaling. Notably, NuRL can raise the model's upper limit, whereas GRPO leaves pass@1024 unchanged from the base model. Furthermore, we present a systematic study of what makes an effective hint and when hints are most useful. Interestingly, the best hints are abstract and high-level–as revealing gold answers actually hurt performance–and are most beneficial when applied necessarily and after GRPO has converged.[2]

## 1 Introduction

Recent advances in reinforcement learning (RL) algorithms have played a central role in improving the reasoning abilities of large language models (LLMs). Despite many promising advances, current online RL algorithms share a key limitation: they cannot learn from problems that are unsolvable under the base policy. In other words, if the model cannot reach the correct answer even after extensive exploration, then no meaningful learning signal can be obtained from the problem. On a similar vein, a growing body of work finds that post-training mainly encourages models to generate already high-reward trajectories (He et al., 2025; Yue et al., 2025; Dang et al., 2025; Zhao et al., 2025). As a result, the model's upper limit–often measured by pass@$k$ for large $k$—remains unchanged after RL training. Intuitively, learning from harder samples offers a clear path to improving a model's performance and expanding its ceiling capacity. In other words, learning from the hard samples has two key benefits: (1) extracting more training signal from the same dataset (improving pass@1),

---

[1]Pronounced like "neural" (nur·uhl)

[2]Code: https://github.com/SalesforceAIResearch/NuRL

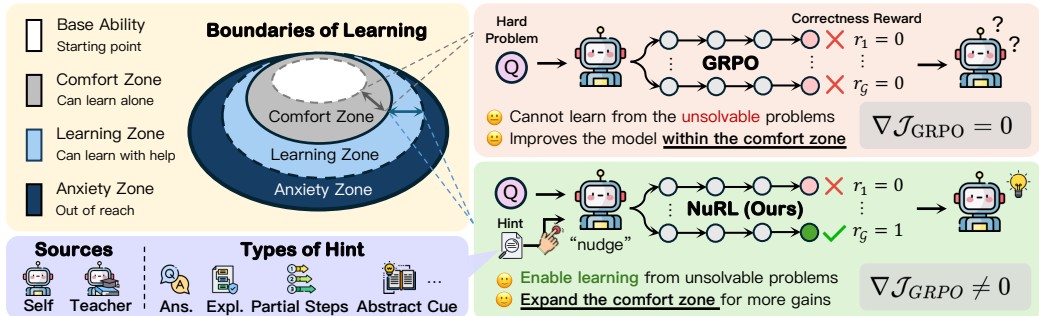

Figure 1: GRPO yields substantial gains, but the improvements largely stem from extending the model's ability within its comfort zone, i.e., if the model fails to solve a hard problem after numerous attempts, it is unable to learn from that problem. In NuRL, we address this by exploring various forms of hints (abstract cues, partial steps, explanations, or even the gold answer), which can be self-generated or teacher-generated. Both self- and teacher-generated abstract cues can expand the model's comfort zone, effectively transforming previously unsolvable problems into solvable ones.

and (2) enabling the model to solve previously unsolvable problems, thereby extending its capability boundary (improving pass@$k$). However, precisely because these problems are difficult, models often cannot learn them without appropriate guidance or intervention from a teacher model.

This dynamic parallels Vygotsky's concept of the Zone of Proximal Development (ZPD) (Vygotsky et al., 1978), which distinguishes between tasks a learner can solve independently ("comfort zone" in Fig. 1) and those achievable only with appropriate guidance ("learning zone" in Fig. 1). The inability to learn from hard samples – or the lack of improvement in pass@$k$ – mirrors being trapped in the comfort zone. Motivated by this analogy, we ask: *can models generate their own "hints""—lightweight forms of guidance—so that even the hard problems become learnable?* Then, we propose NuRL (**Nu**dging LLM with **R**einforcement **L**earning), which adaptively injects self-generated hints into training. Our hypothesis is that hard problems become more learnable when paired with carefully abstracted hints (Huang et al., 2025; Park et al., 2025). These hints act as light-weight "nudges," transforming previously unlearnable samples into productive training signals. To achieve this, our approach begins with offline hint collection. Given a question and its gold answer, we prompt the model to generate Chain-of-Thought (CoT; Wei et al., 2022) reasoning that connects the two. Using the question and CoT as input, the model then produces a high-level cue that captures the core knowledge required to solve the problem. We also explore various types of hints as shown in the bottom-left of Fig. 1, where the hints can be self-generated or provided by a stronger model (e.g., GPT-o4-mini (OpenAI, 2025)), and the forms of hints can be abstract cues, partial steps, explanations, or even the gold answer. We find that self-generated hints are effective, while teacher-generated hints give further improvements. Importantly, effective hints are abstract and conceptual: they neither reveal the final answer nor provide detailed solution steps, but only mention what core knowledge is needed to solve this problem (see the bottom left in Fig. 2).

We adopt GRPO (Shao et al., 2024) as the training framework. During training, the policy model generates $\mathcal{G}$ rollouts per problem, and we use pass rate to decide when to inject hints. For hard problems with a 0% pass rate, we inject the pre-generated hints to the end of the question, and prompt the model to regenerate another $\mathcal{G}$ rollouts conditioned on the hints. Given the hint, the model is more likely to produce successful solutions (non-zero pass rate), turning previously unlearnable examples into learnable ones. This strategy offers two main advantages: (1) Hints boost pass rate, enabling hard problems to produce meaningful training signals. (2) Hints remain within the model's distribution, since they are self-generated (conditioned on the answer), avoiding distributional shift and does not require stronger external models. Together, these benefits allow NuRL to unlock value from harder samples, broadening the set of problems that contribute to RL training.

We evaluate NuRL on six diverse benchmarks across multiple domains, including MATH 500, MATH Hard, AIME, GPQA, MMLU-Pro and Date Understanding. Results show that NuRL boosts the average performance with three different models (+1.62% using Llama, +1.75% using Octo-Thinker, and +0.79% using Qwen as compared to GRPO), and when an external teacher model is available for hint generation, the improvement can be further enlarged to up to 3.44%. More-

over, our analysis shows that (a) NuRL is complementary to test-time scaling methods such as Self-Consistency (Wang et al., 2022) and shows larger improvement: our method improves 9.4% with 16-way Self-Consistency, as compared to GRPO, which improves 7.8%. (b) NuRL is able to transform previously unsolvable problems into solvable ones, and that transfer to the improvements on the upper limit of a model's capacity. (c) Hints are useful when they are abstract and high-level. *The more exposure to the answer, the more severe the degradation.* This aligns with human learning, where effective hints should be abstract and high-level—providing guidance without revealing the solution. Knowing the answer upfront risks biasing toward it and undermines generalization.

## 2 RELATED WORK

**Reinforcement Learning with Verifiable Reward (RLVR).** RLVR computes the reward using rule-based verification, which is effective in improving LLM reasoning. The reward function can be as simple as checking whether the model's answer matches the gold answer (Lambert et al., 2024; Guo et al., 2025; Team et al., 2025; Zeng et al., 2025). The success of RLVR is also supported by advancements, including PPO (Schulman et al., 2017), DPO (Rafailov et al., 2023), GRPO (Guo et al., 2025) and many techniques like DAPO (Yu et al., 2025), Dr. GRPO (Liu et al., 2025c). Recent work explores different strategies to learn a better self-reflection ability, such as "try again" (Liu et al., 2025a) or self-critique (Zhang et al., 2025c).

**The Role of RL: Distribution Sharpening vs. Discovery.** There has been an active discussion on whether RL primarily performs *distribution sharpening*, i.e., amplifying behaviors already present in the model, or enables genuine discovery of new reasoning abilities. The distribution-sharpening view holds that RL mainly surfaces high-reward paths and increases their likelihood of generation (Zhang et al., 2025a; Zhao et al., 2025; Shenfeld et al., 2025). This is often supported by the findings that RL improves pass@1 but not pass@$k$ (Yue et al., 2025; He et al., 2025), and that even weak reward signals can yield substantial improvements (Shao et al., 2025; Zuo et al., 2025; Prabhudesai et al., 2025; Wang et al., 2025b). On the other hand, some argue that RL fails to improve pass@$k$ not because of inherent limitations, but due to insufficient training or evaluating on tasks where models already perform strongly (Liu et al., 2025b). Yuan et al. (2025) also show that RL can learn new skills by combining existing ones. Zhang et al. (2025a) suggest that RL can both sharpen and discover, with the balance determined by the trade-off between exploration and exploitation.

**Mixture of Offline and Online RL.** When the base policy fails to generate solutions that yield non-zero rewards, replay buffer or off-policy optimization leverage previous positive trajectories (Lu et al., 2025) or expert demonstrations (Levine et al., 2020). Recently, hybrid approaches that mix online and offline training have been proposed to improve generalization (Yan et al., 2025; Phan et al., 2025). TBA (Bartoldson et al., 2025) decouples data generation from policy updates by leveraging multiple searcher nodes to produce diverse trajectories, enabling high resource utilization and a more scalable search process. A line of work uses SFT with RL to expand the model's knowledge scope (Ma et al., 2025; Fu et al., 2025; Zhou et al., 2025), while others employ hints (e.g., partial solutions) to adjust problem difficulty during training (Zhang et al., 2025b; Huang et al., 2025; Li et al., 2025; Qu et al., 2025; Wang et al., 2025a). Our work differs from these approaches in two aspects. First, NuRL does not rely on SFT to broaden the model's knowledge; instead, we focus on the RL stage and analyze how different hints expand a model's reasoning boundary. A notable work is STaR (Zelikman et al., 2022), where the reasoning is bootstrapped from reasoning given the answer. Our hints are further abstracted out from the reasoning to ensure they do not disclose the answer. Second, we show that nudging the model with self-generated hints is effective: it enables self-improvement (both pass@1 and pass@$k$) without depending on stronger external models.

## 3 METHODOLOGY

### 3.1 PRELIMINARY

Our method is based on GRPO (Shao et al., 2024). Specifically, GRPO updates the policy by maximizing $\mathcal{J}_{\text{GRPO}}(\theta)$ using the following objective:

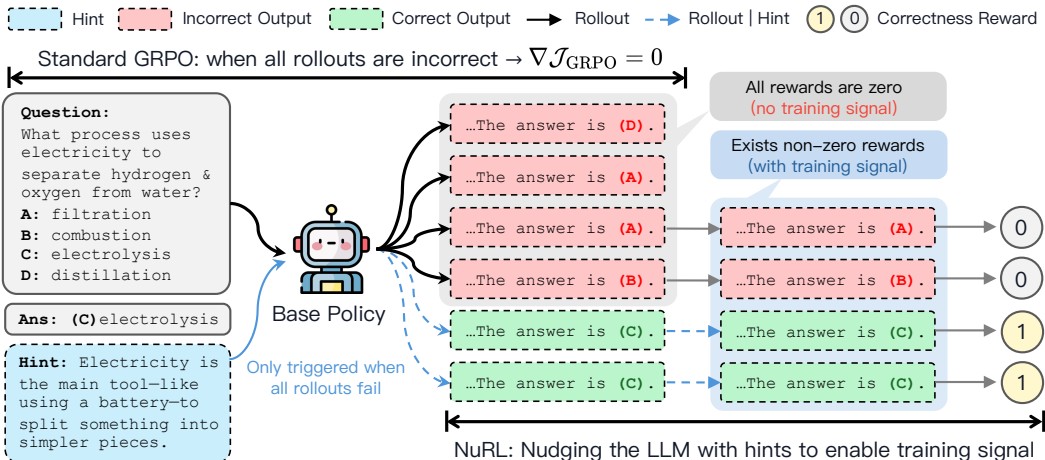

Figure 2: NuRL provides targeted guidance to the LLM policy during online GRPO training. Prior to training, we construct an offline collection of hints, defined as abstract problem-specific cues that reduce task difficulty. During the online training, whenever all $\mathcal{G}$ rollouts for a problem are incorrect, NuRL augments $\mathcal{G} - 1$ of the rollouts with the corresponding hint and regenerates the batch. This intervention facilitates the acquisition of non-zero rewards on instances that would otherwise yield uniformly zero rewards, thereby supplying informative training signals.

$$\mathcal{J}_{\text{GRPO}}(\theta) = \frac{1}{\mathcal{G}} \sum_{i=1}^{\mathcal{G}} \frac{1}{|o_i|} \sum_{t=1}^{|o_i|} \min \left[ \frac{\pi_\theta(o_{i,t}|o_{i,<t})}{\pi_{\text{old}}(o_{i,t}|o_{i,<t})} \hat{A}_{i,t}, \text{clip}\left( \frac{\pi_\theta(o_{i,t}|o_{i,<t})}{\pi_{\text{old}}(o_{i,t}|o_{i,<t})}, 1 - \varepsilon, 1 + \varepsilon \right) \hat{A}_{i,t} \right],$$

where $\pi_\theta$ is the policy, $\pi_{\text{old}}$ is the old policy, $\varepsilon$ is the clipping range, and $\hat{A}_t$ is an estimator of the advantage at time step $t$. Given a reward function $f$ and a question-answer pair $(q, a)$ from training data $\mathcal{D}$, the advantage is estimated by letting $\pi_{\theta_{\text{old}}}$ samples a group of $\mathcal{G}$ responses $\{o_i\}_{i=1}^{\mathcal{G}}$. Then, the advantage of the $i$-th response is calculated by normalizing the rewards within the group $\{r_i\}_{i=1}^{\mathcal{G}}$:

$$\hat{A}_{i,t} = \frac{r_i - \mu_r}{\sigma_r}, \quad \mu_r = \frac{1}{\mathcal{G}} \sum_{i=1}^{\mathcal{G}} r_i, \quad \sigma_r = \sqrt{\frac{1}{\mathcal{G}} \sum_{i=1}^{\mathcal{G}} (r_i - \mu_r)^2}. \tag{1}$$

We use the rule-based outcome correctness as the reward (Guo et al., 2025), i.e., $f(\hat{y}, y) = 1$, if `is_equivalent`$(\hat{y}, y)$ else 0, where $y$ is the ground truth and $\hat{y}$ is the predicted answer.

## 3.2 NuRL: Nudging LLMs with Reinforcement Learning

**Offline Hint Collection.** Given a training dataset consisting of question–answer pairs $\mathcal{D} = (q_i, a_i)_{i=1}^{N}$, our method begins with offline hint collection. As illustrated in Fig. 1, hints can be categorized by their *source* (self-generated by the model vs. provided by a teacher model) and their *type*: **(1) Abstract cues**: generated by abstracting from an explanation of why an answer is correct, designed to avoid revealing concrete details or the final answer. **(2) Partial steps**: obtained by generating a step-by-step solution using the gold answer, masking 75% of the steps (keep only the first 25%), with the model asked to complete the rest. **(3) Explanations**: formed by replacing incorrect rollouts with gold-conditioned explanations that justify why the answer is correct. **(4) Ground-truth answer**: appending the correct answer alongside the question with the prompt, "I was told the answer is {gold_answer}. Do not blindly accept it. Take it as a reference and provide your own step-by-step reasoning." Note that from (1) to (4) is a reverse order of how much information about the answer is being revealed. Later in Section 5.2, we will show that the more disclosure about the answer, the worse the performance is. Therefore, in this work, we mainly focus on **self-generated abstract cues** as the hint (see Appendix E.1 for examples with different types of hints).

Concretely, to collect such high-level abstract cues that can simplify the problem without revealing specific solution details, we first prompt the base policy LLM with both the question and the ground-truth answer, and instruct it to generate a Chain-of-Thought (CoT; Wei et al., 2022) that explains why the provided answer is correct. This can be expressed as $y = \pi_{\text{old}}(q, a; p_y)$ where $\pi_{\text{old}}$ is the initial policy, $y$ is the generated CoT, and $p_y$ is the prompt (see Appendix D). We find that directly using such explanation-style CoTs as hints is not effective (also will be shown in Section 5.2). To address this, we introduce an abstraction step. Specifically, we prompt the LLM again with the question, the answer, and the self-generated CoT $(q, a, y)$, asking it to produce a high-level hint that reduces task difficulty without being overly specific: $h = \pi_\theta(q, a, y; p_h)$ where $p_h$ is the hint generation prompt as provided in Appendix D. After this process, the training data is augmented with hints, yielding an enriched dataset $\mathcal{D} = \{(q_i, a_i, h_i)\}_{i=1}^N$ where every sample has a corresponding hint.

**Online Rollout Augmentation.** After augmenting the training data with hints, we proceed with online GRPO-style training. For each question $q$, the base policy first generates $\mathcal{G}$ rollouts without hints. When all responses $\{o_i\}_{i=1}^{\mathcal{G}}$ are incorrect (i.e., $r_i = 0 \Rightarrow \hat{A}_{i,t} = 0$ for all $i$), the advantages vanish, yielding $\nabla \mathcal{J}_{\text{GRPO}} = 0$ and thus no policy update. This is illustrated in the gray box of Fig. 2, where the correct answer is C but none of the rollouts reach it. Similarly, if all rollouts are correct, the task is trivially easy and again results in $\nabla \mathcal{J}_{\text{GRPO}} = 0$. In practice, such uninformative problems (both too hard and too easy) are often discarded to improve training efficiency (Yu et al., 2025). In contrast, NuRL retains the hard cases, as they present opportunities to unlock further gains. Specifically, when all $\mathcal{G}$ rollouts fail, we activate NuRL by appending the offline-generated hint $h$ to the problem, producing the new input $q \oplus h$, where $\oplus$ denotes concatenation. A new batch of rollouts is then generated from $q \oplus h$ (blue box in Fig. 2). To reduce the chance to collapse into uniformly correct responses—which would again eliminate the learning signal—we let only $\mathcal{G} - 1$ rollouts see the hint, $\{o_i\}_{i=1}^{\mathcal{G}-1} \sim \pi_{\theta_{\text{old}}}(q, h)$, while one rollout remains hint-free, $o_\mathcal{G} \sim \pi_{\theta_{\text{old}}}(q)$.

**Inference.** The hints are only used in training. During test time, we prompt the model only with the question. The hypothesis is that exposure to hints during training guides the model toward correct solutions, enabling it to internalize the reasoning patterns required to solve the problems. That is, the use of hints broadens the set of solvable problems and translates to improved performance.

## 4 EXPERIMENTAL SETUP

**Models.** We evaluate NuRL on three models: Llama3.2-3B-Instruct (Grattafiori et al., 2024), OctoThinker-3B-Hybrid-Zero (Wang et al., 2025c), and Qwen3-4B-Instruct-2507 (Team, 2025).[3] Llama is a general-purpose instruction-tuned model, while OctoThinker is a recently proposed model that applies *mid-training* on Llama and has been shown to exhibit stronger compatibility with reinforcement learning (Wang et al., 2025c). Qwen is also a general-purpose instruction-tuned model that encompasses strong compatibility with post-training techniques. We evaluate these models because they exhibit distinct properties, helping us avoid conclusions that are overly specific to a single model—a lesson underscored in recent studies (Shao et al., 2025; Chandak et al., 2025).

**Datasets.** For training, we use Open-R1's Mixture-of-Thought dataset (Face, 2025) due to its diversity, which includes science QA data from Llama-Nemotron (Bercovich et al., 2025), ensuring our training set is not purely math-focused. We randomly sample 7.5k math and 2.5k science data points. Since the dataset provides only CoT outputs from Deepseek-R1 (Guo et al., 2025) rather than explicit gold answers, we extract the answers from \boxed{} and generate 8 CoTs per problem using GPT-o4-mini (OpenAI, 2025). We keep only samples where Deepseek-R1 and GPT-o4-mini's majority agree on the answer. This yields 8.3k samples for training. NuRL is tested across a diverse set of reasoning-intensive tasks spanning STEM and other domain-specific areas, including: **(1) MATH 500**, a subset 500 problems from the MATH benchmark (Hendrycks et al., 2021) curated by Lightman et al. (2023); **(2) MATH Hard**, the hardest problem set from MATH, totaling 1.3k problems (Hendrycks et al., 2021); **(3) AIME 2024**, 30 mathematics problems from the 2024 AIME competition (AIME, 2024); **(4) GPQA Diamond**, 198 PhD-level questions covering biology, physics, and chemistry (Rein et al., 2024); **(5) MMLU-Pro**, a more challenging variant of MMLU (Hendrycks et al., 2020), spanning 14 college-level subjects with 12k samples (Wang et al., 2024); **(6) Date Understanding**, 250 problems designed to test LLM's understanding to date information, requiring commonsense and logical reasoning (bench authors, 2023; Suzgun et al., 2022).

---

[3]Hereafter, we refer to them as Llama, OctoThinker, and Qwen, respectively.

Table 1: Comparison of methods across three models and six benchmarks. NuRL consistently outperforms all baselines with self-generated hints, and shows further improvements when an external model is available for hint generation (shown in gray for reference).

| | MATH 500 | MATH Hard | AIME | GPQA | MMLU-Pro | Date | Avg. |
|---|---|---|---|---|---|---|---|
| **Llama3.2-3B-Instruct** | | | | | | | |
| Zero-shot | 35.71 | 15.28 | 3.33 | 13.23 | 11.52 | 2.33 | 13.57 |
| Few-shot | 36.68 | 16.73 | 3.63 | 15.52 | 12.02 | 8.87 | 15.58 |
| RFT | 40.12 | 17.72 | 3.63 | 14.76 | 12.98 | 13.36 | 17.10 |
| ReFT | 55.80 | 28.86 | 8.00 | 24.42 | 30.19 | 55.34 | 33.77 |
| GRPO | 56.92 | 30.11 | 8.33 | 27.98 | 34.78 | 57.10 | 35.87 |
| w/ Hint (Self) | **58.04** | **31.62** | **9.17** | **28.28** | **36.18** | **61.65** | **37.49** |
| w/ Hint (GPT-o4-mini) | 59.30 | 32.42 | 10.83 | 28.40 | 41.38 | 63.53 | 39.31 |
| **OctoThinker-3B-Hybrid-Zero** | | | | | | | |
| Zero-shot | 59.98 | 36.87 | 4.83 | 16.57 | 21.18 | 19.25 | 26.45 |
| Few-shot | 61.23 | 37.01 | 5.33 | 18.62 | 23.32 | 24.53 | 28.34 |
| RFT | 62.78 | 37.73 | 4.83 | 19.45 | 27.34 | 36.82 | 31.49 |
| ReFT | 66.38 | 39.69 | 6.66 | 24.53 | 44.66 | 70.32 | 42.04 |
| GRPO | 68.81 | 41.29 | 8.33 | 23.26 | 44.25 | 69.85 | 42.63 |
| w/ Hint (Self) | **70.13** | **42.07** | **9.66** | **27.15** | **45.54** | **71.75** | **44.38** |
| w/ Hint (GPT-o4-mini) | 71.62 | 43.51 | 12.63 | 27.43 | 46.53 | 72.28 | 45.67 |
| **Qwen3-4B-Instruct-2507** | | | | | | | |
| Zero-shot | 94.88 | 90.54 | 58.75 | 35.57 | 58.88 | 83.35 | 70.33 |
| Few-shot | 94.97 | 90.54 | 55.52 | 34.82 | 59.01 | 85.01 | 69.98 |
| RFT | 94.41 | 90.41 | 56.63 | 37.72 | 59.54 | 85.42 | 70.69 |
| ReFT | 96.46 | 90.83 | 62.79 | 60.31 | 72.21 | 92.20 | 79.13 |
| GRPO | **96.52** | 90.54 | 60.83 | 62.50 | 72.65 | **92.80** | 79.31 |
| w/ Hint (Self) | 96.46 | **92.57** | **63.54** | **62.88** | **72.83** | 92.30 | **80.10** |
| w/ Hint (GPT-o4-mini) | 96.58 | 92.96 | 62.71 | 64.99 | 72.95 | 93.91 | 80.68 |

**Baselines.** We compare NuRL with the following baselines: (1) **Zero-shot:** We prompt the model to think step-by-step and provide the answer within \boxed{} (Kojima et al., 2022). (2) **Few-shot:** Besides prompting the model to think step-by-step, we include 8 in-context learning samples. (3) **Rejection sampling Fine-Tuning (RFT):** We prompt the model with training data 8 times using the zero-shot prompt same as above, and keep only the correct reasoning chains to perform supervise fine-tuning (Yuan et al., 2023). (4) **Reasoning with Reinforced Fine-Tuning (ReFT):** We adopt the SFTed model from RFT, and continue for GRPO training (Trung et al., 2024). (5) **GRPO**: Using outcome correctness as a rule-based reward function (Guo et al., 2025).

**Implementation Details.** We employ abstract cues as hints, and compare self-generated versus teacher-generated in Table 1. We evaluate all methods using pass@1. Results are averaged over 16 runs, except for MMLU-Pro, which has 12k samples; for this dataset, we report the average over 3 runs. We adopt verl (Sheng et al., 2024) as the backbone, and utilize vllm (Kwon et al., 2023) to speed up rollout generation and inference. To verify equivalence between predictions and references, we use Math-Verify[4]. We employ a two-stage strategy for training. In stage 1, we optimize the base policy with correctness-only GRPO until both training reward and validation accuracy show no improvement for over 10 steps. In stage 2, we apply NuRL to continue training. To ensure fairness, GRPO-based baselines are trained for the same total number of steps as NuRL, albeit NuRL starts midway. Before stage 2 begins, we use the stage 1 checkpoint to generate 8 rollouts and filter out samples where all rollouts are correct (i.e., overly easy cases) to improve efficiency. The resulting sample size is reported in Appendix B. All methods are running on 8 H200 GPUs, and it takes around six days for GRPO-based methods to converge. During training, we use a temperature of 1.0

---

[4] https://github.com/huggingface/Math-Verify

and set the clip-high parameter $\epsilon$ to 0.28 (Yu et al., 2025). The rollout number is 16 for GRPO and 8 for NuRL, as NuRL may regenerate an additional batch of rollouts when all rollouts fail. We cap the output length at 9k tokens for both training and testing. At inference, we fix the temperature to 0.7. Token limits and inference temperature are aligned across all baselines for evaluation. Other hyperparameters and details can be found in Appendix B and in our code.

## 5 RESULTS AND ANALYSIS

### 5.1 MAIN RESULTS

**NuRL consistently outperforms all baselines with self-generated hints.** We present the main results in Table 1. Across six benchmarks, NuRL shows superior performance compared to all baselines, beating the zero-shot and the SFT baseline (RFT) by a large margin. For RL-based baselines (ReFT and GRPO), we see the performance increase by a large margin (+8.8% to +22.3% compared to the base model), conforming that RL largely improves a model's performance. Nevertheless, NuRL consistently surpasses these strong RL-based baselines. On average, it improves over GRPO by +1.62 points on Llama, +1.75% on OctoThinker, and +0.79% on Qwen. Importantly, these improvements are achieved on top of already strong GRPO performance, despite GRPO using 16 rollouts per question, whereas NuRL uses only 8 (with the option to generate another eight only if all initial rollouts fail). Thus, GRPO operates with strictly more rollouts than NuRL. Since the primary difference between NuRL and GRPO is the use of hints, these results highlight that targeted hints are an effective mechanism for improving LLM performance. We note that the relatively smaller improvement on Qwen (+0.79%) may stem from the limited stage 2 data (fewer than 2k examples), as Qwen's stronger base capability caused many overly easy samples to be filtered out before stage 2. Finally, we observe that incorporating stronger external models for hint generation yields additional gains beyond self-generated hints. For example, on Llama, hints from GPT-o4-mini improve performance by +3.44% absolute points on average over GRPO and by +1.82% over self-hints. This demonstrates that while self-generated hints are already beneficial, the framework naturally accommodates stronger sources of guidance when available.

### 5.2 ADDITIONAL ANALYSIS

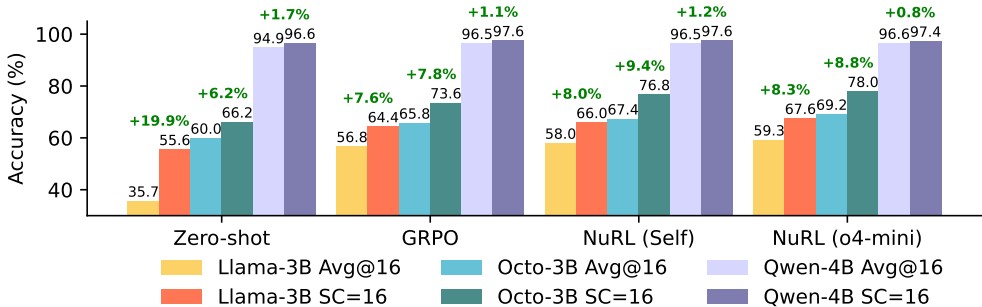

Figure 3: Compared to GRPO's improvements with Self-Consistency (+7.6%, +7.8% and +1.1% on Llama, OctoThinker and Qwen), NuRL obtains larger gains with +8.0%, +9.4% and 1.2%.

**NuRL is complementary to test-time scaling method.** While NuRL shows superior performance in Table 1, a common way to augment the baselines is to scale the test-time compute, often measured in the number of tokens or number of samples at inference time. Here we adopt the latter and employ Self-Consistency (SC; Wang et al., 2022). We compare NuRL with zero-shot and GRPO, with and without Self-Consistency across the three models in MATH 500. Results in Fig. 3 show that NuRL not only remains effective for test-time scaling method, but also shows greater improvements compared to GRPO. Specifically, Llama trained with GRPO improves 7.6% with SC, while Llama trained with NuRL improves 8.0%. Similarly, OctoThinker trained with GRPO improves 7.8% with SC, while OctoThinker trained with NuRL improves 9.4%. Finally, Qwen trained with GRPO improves 1.1% with SC, while Qwen trained with NuRL improves 1.2%.

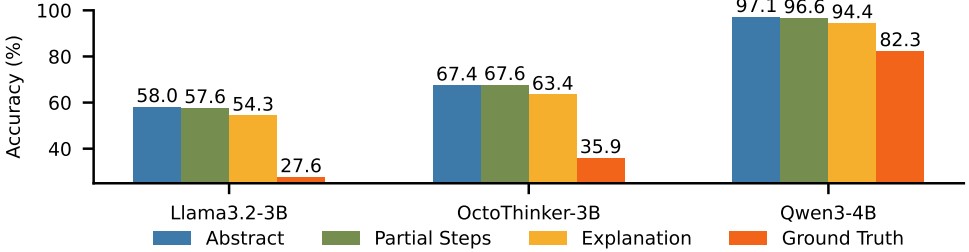

Figure 4: Comparison of different types of hints. From left to right, the hints vary in how directly they disclose information about the ground-truth answer. At the leftmost end, abstract hints provide only high-level guidance without revealing details of the solution or answer, whereas at the rightmost end, the answer is given explicitly. Interestingly, more direct hints lead to worse performance.

**Hint abstraction is key to improvement.** In Section 3, we described how hints are generated by abstracting explanations of why an answer is correct. While this represents one useful approach, it raises a broader question: *what makes a good hint?* To investigate this, we compare four types of hints as mentioned in Section 3. In Fig. 4, we observe a consistent trend: the more directly answer information is disclosed, the lower the downstream performance. Abstract cues, which explicitly avoid revealing details or solutions, yield the highest accuracy. Partial steps perform slightly worse, but still help since the initial reasoning structure provides a useful starting point. Explanations are less effective, as justifying why an answer is correct is not equivalent to reasoning through the problem; in fact, explanations often implicitly disclose the answer (e.g., "*but the answer is {gold_answer}, so I should try a different method*"). Finally, directly providing the ground-truth answer severely harms generalization. This setup frequently induces reward hacking: during training, the model learns to simply output the provided answer to maximize reward without genuine reasoning, and that does not generalize at test time. Overall, these results suggest that effective hints should remain high-level and abstract. A good hint guides the model toward reasoning independently, but avoids revealing shortcuts that undermine generalization.

**Using hints only when necessary is crucial.** While we have established that a good hint should be abstract, the question of *when is the best time to use hints* remains open. To examine this, we consider two key factors. First, one can decide whether to apply hints from the beginning of training, or to wait until GRPO converges before introducing them. We refer to the latter approach as *two-stage training*. Second, one can determine the condition under which hints are provided. Specifically, hints may be applied uni-

Table 2: Comparison of hint application strategies during training. Results are on MATH 500 and GPQA with Llama3.2-3B-Instruct.

| Two-stage | Diff. Trigger | MATH | GPQA |
|:---:|:---:|:---:|:---:|
| ✗ | ✓ | 56.06 | 27.63 |
| ✗ | ✗ | 53.41 | 24.84 |
| ✓ | ✗ | 53.09 | 26.62 |
| **✓** | **✓** | **58.04** | **28.28** |

formly to all problems, or only when all $\mathcal{G}$ rollouts are incorrect. We refer to this latter condition as a *difficulty trigger*. Combining these two factors yields four experimental settings, summarized in Table 2. We find that applying hints from the beginning of training generally underperforms compared to introducing hints only after GRPO has stabilized. Similarly, applying hints indiscriminately results in lower performance than using a difficulty trigger. This suggests that unnecessary hints may interfere with the model's ability to learn independently on problems where guidance is not required. In contrast, when hints are introduced only after GRPO convergence and combined with a difficulty trigger, we observe consistent improvements across both MATH and GPQA (58.04 and 28.28, respectively). In summary, these findings indicate that hints are most effective when used selectively and adaptively. Rather than being injected throughout training or applied uniformly, hints should be reserved for difficult cases and integrated after the base policy has stabilized.

**NuRL improves pass@k when the task is more challenging to the model.** In Table 1, we show that NuRL generally outperforms GRPO in terms of pass@1. A natural follow-up question is whether these gains also extend to higher values of $k$, especially when using NuRL, the model is supposed to solve more training problems compared to GPRO. To investigate this, we plot pass@$k$ for $k = \{1, 2, \ldots, 512, 1024\}$ in Fig. 5 using Llama. Following Yuan et al. (2025), we set the maxi-

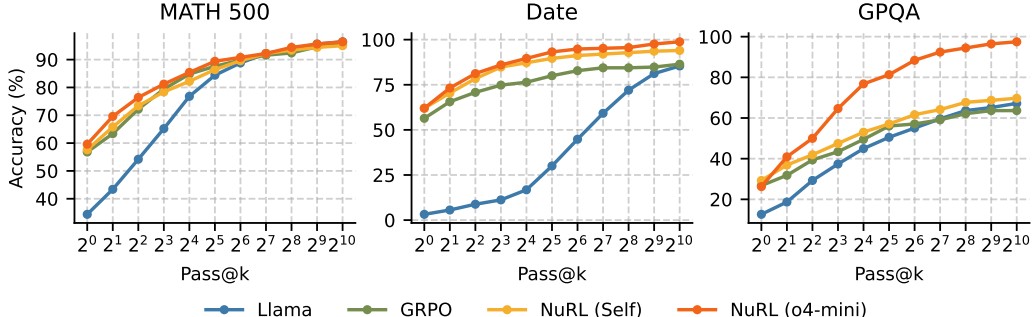

Figure 5: When the base model (Llama) already has strong pre-trained knowledge (e.g., MATH 500), both GRPO and NuRL yield little improvement in pass@k. In contrast, on tasks with lower upper-bound performance (e.g., Date Understanding and GPQA, with pass@1024 of 85.4 and 67.2), GRPO provides no gains on pass@1024, while NuRL pushes it further.

mum $k$ as 1024 as a sufficiently large and practical budget to probe the model's ceiling performance. In Fig. 5, we first report results on MATH 500. Here, the base model already achieves pass@1024 = 96.4% before training (despite its relatively low pass@1 = 34.4%). In this case, neither GRPO nor NuRL increases pass@1024, as the model already possesses strong knowledge on this task, albeit hard to generate a correct answer given only one attempt. In contrast, for tasks where the base model's upper bound is lower – such as Date Understanding (85.4%) and GPQA (67.2%) – we observe clear gains from NuRL, while GRPO provides little or no improvement. We hypothesize that hints guide the model to explore underrepresented solution paths, increasing the chance of discovering correct answers that would otherwise remain in "unreachable corners" of the search space. From an entropy perspective, hints may also induce more diverse exploration, which translates into higher pass@$k$ (Cheng et al., 2025). Importantly, *these hints are lightweight*: rather than fine-tuning the model with hints, we simply append them to the input question, yet this is sufficient to guide exploration more effectively. Moreover, we find that pass@1024 scales with the quality of hints: using teacher-generated hints pushes it further to 95.2% on GPQA, compared to GRPO's 63.4%. In summary, when the base model's ceiling performance is not yet saturated, NuRL raises pass@$k$ – both with self-generated hints (86.4% → 94.0% on Date, 63.6% → 69.7% on GPQA) and even more so with higher-quality hints generated from the teacher model.

**NuRL increases the fraction of solvable problems.** Having demonstrated that NuRL is able to improve both pass@1 and pass@$k$, we further analyze the source of such performance gains. In Fig. 6, we show that training with self-generated hints effectively increases the fraction of solvable problems in the training data. Specifically, we present Qwen's training log and compare, at each training step, the fraction of solvable problems under three conditions: (1) no hints, corresponding to standard GRPO training, (2) using NuRL before adding hints, and (3) using NuRL after injecting hints whenever all rollouts fail. Here, a problem is considered solvable if at least one rollout produces a correct answer. Given the same training step, adding hints yields an approximate 4% increase

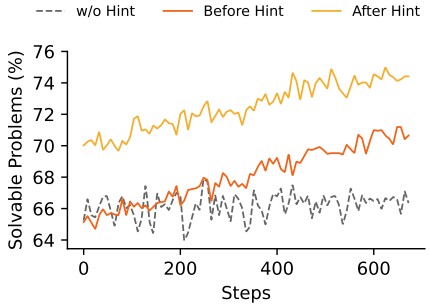

Figure 6: The self-generated hints in NuRL effectively reduce the task difficulty and increase the portion of solvable problems.

in solvable problems, suggesting that self-generated hints reduce problem difficulty for the model. Second, NuRL shows a clear upward trend in the fraction of solvable problems both before and after hint injection. This indicates that as training progresses, the model begins to solve problems that were previously unsolvable, increasing the solvable fraction from 66% to 70%. Lastly, standard GRPO without hints exhibits a relatively flat trend, with the solvable fraction fluctuating around 66%. Overall, these results explain the improvements observed across diverse benchmarks: by leveraging self-generated hints, the model effectively increases the amount of data it can learn from.

## 6 CONCLUSION

We introduce NuRL, a self-guided reinforcement learning approach that uses hints to extend models' reasoning capabilities. NuRL consistently outperforms strong baselines with self-generated hints and achieves further gains with hints generated by a stronger external model. Our analysis reveals that the most effective hints are high-level, abstract, and applied selectively – only when the model cannot solve a problem unaided. Moreover, NuRL scales more efficiently than GRPO at test time, increases the fraction of solvable problems, and delivers substantial pass@$k$ improvements on two challenging tasks, effectively expanding the model's comfort zone.

## ETHICS STATEMENT

In this work, we propose a reinforcement learning method that optimizes for final outcome correctness. As a result, our trained LLMs may still produce hallucinations, since intermediate reasoning is not directly supervised and the correctness is only verified against the final answer. Thus, outputs generated by NuRL carry potential risks of misinformation or hallucination. Future research is needed to better assess and mitigate these limitations.

## REPRODUCIBILITY STATEMENT

We are making our code available in the supplementary materials to enable replication of our findings. We also provide implementation details of NuRL in Appendix B and prompts in Appendix D. The datasets we use are all publicly available, as detailed in Appendix C.

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

## A    THE USE OF LARGE LANGUAGE MODELS (LLMS)

We use ChatGPT[5] for grammar correction and refinement. The model was only used to polish text already written by the authors, and was not used for research ideation or generating original content.

---

[5]https://chatgpt.com/

# B    IMPLEMENTATION DETAILS

Recall that in Section 3, we describe a two-stage training procedure where NuRL is applied after GRPO converges. By convergence, we mean that both the training reward and validation accuracy plateau after 10 training steps. Stage 1 corresponds to GRPO convergence, and stage 2 applies NuRL on top of the converged checkpoint. For fairness, GRPO-based baselines also undergo stage 2 training without hints, ensuring the total training steps are aligned with NuRL. Since convergence speed varies across models, we report the detailed hyperparameters and number of steps required for each model in Table 3.

Table 3: **Stage 1** configurations and hyperparameters.

|  | **Llama** | **OctoThinker** | **Qwen** |
|---|---|---|---|
| num_train_samples | 8316 | 8316 | 8316 |
| max_prompt_length | 1800 | 1800 | 1800 |
| max_response_length | 9000 | 9000 | 9000 |
| lr | 1e-6 | 1e-6 | 1e-6 |
| clip_ratio_low | 0.2 | 0.2 | 0.2 |
| clip_ratio_high | 0.28 | 0.28 | 0.28 |
| rollout_temperature | 1 | 1 | 1 |
| rollout_n | 16 | 16 | 16 |
| use_kl_loss | False | False | False |
| train_batch_size | 1024 | 1024 | 4096 |
| converged_steps | 375 | 175 | 125 |

At the start of Stage 2, we use the converged checkpoints and generate 8 rollouts per question. Then, we discard samples where all rollouts are correct (i.e., overly easy examples) to improve efficiency. This filtering is performed after Stage 1, since the trained checkpoints are stronger than the initial models. The detailed stage 2 configurations are given in Table 4.

Table 4: **Stage 2** configurations and hyperparameters.

|  | **Llama** | **OctoThinker** | **Qwen** |
|---|---|---|---|
| num_train_samples | 5996 | 4502 | 1937 |
| max_prompt_length | 1800 | 1800 | 1800 |
| max_response_length | 9000 | 9000 | 9000 |
| lr | 1e-6 | 1e-6 | 1e-6 |
| clip_ratio_low | 0.2 | 0.2 | 0.2 |
| clip_ratio_high | 0.28 | 0.28 | 0.28 |
| rollout_temperature | 1 | 1 | 1 |
| rollout_n | 16 | 16 | 16 |
| use_kl_loss | False | False | False |
| train_batch_size | 512 | 512 | 1937 |
| converged_steps | 275 | 175 | 90 |

# C    DATASET STATISTICS AND LICENSES

We provide the sample sizes and licenses of the datasets used in this work in Table 5. All the datasets are in English and all datasets are used in a fashion consistent with their intended use.

Table 5: The statistics and licenses of the datasets used in this study.

|  | Sample Size | License |
| --- | --- | --- |
| MATH 500 (Lightman et al., 2023) | 500 | MIT License |
| MATH Hard (Hendrycks et al., 2021) | 1324 | MIT License |
| AIME 2024 (AIME, 2024) | 30 | CC0 |
| GPQA-Diamond (Rein et al., 2024) | 198 | MIT License |
| MMLU-Pro (Wang et al., 2024) | 12032 | Apache License |
| Date Understanding (bench authors, 2023; Suzgun et al., 2022) | 250 | Apache License |

## D PROMPTS FOR HINT GENERATION

We use the following prompts to generate hints. First, we generate an explanation style of Chain-of-Thought by conditioning on the ground truth answer:

---

**Prompt for Explanation Generation**

Question: {question}

I was told the answer is {gold_answer} but I don't know why. Please explain why the answer is {gold_answer} step by step.

---

Then, we use the concatenation of the question with the generated explanation, and ask the model to generate an abstract hint using the following prompt:

---

**Prompt for Hint Generation**

<system_prompt> You are a tutor. You are given a set of question, correct answer and solution. Your job is to provide a hint for the problem. The hint should help the student learn the core concept (e.g. formula, lemma, or necessary knowledge) needed to solve this problem. The hint should be concise, to the point, but high level. Do not include any detailed steps or calculations or the final answer. </system_prompt>

Question: {question}

Answer: {gold_answer}

Solution: {explanation}

Now, please provide a hint for this problem to help the student learn the core concept.

---

# E  QUALITATIVE EXAMPLES

## E.1  EXAMPLES OF DIFFERENT TYPES OF HINTS

We provide examples of the four types of hints we explored in this work. All the examples below answer the same question (shown below) and are self-generated with Qwen3-4B-Instruct-2507.

---

**Example Question and Answer**

**Question:** During a school football match, there are 3 types of balls used, each with a different number of colors. Type A balls have yellow, brown, and blue variations; Type B balls have green, orange, and pink; and Type C balls have black, red, and white variations. The number of colored balls varies according to the number of players in each team. Team 1 has 72 members, and each player receives 4 Type A yellow balls, 3 Type A brown balls, and 3 Type A blue balls. Team 2 has 64 members, and each player receives 5 Type B green balls, 4 Type B orange balls, and 2 Type B pink balls. Team 3 has 53 members, and each player receives 3 Type C black balls, 2 Type C red balls, and 6 Type C white balls. Calculate what percentage of all the balls distributed for the match are Type A yellow balls.

**Answer:** 14.35%

---

**Type 1: Abstract Cue**

To find the percentage of a specific type of ball among all distributed balls, first calculate the total number of that specific ball and then divide it by the total number of all balls distributed—then multiply by 100. Focus on identifying which team(s) contribute to the specific ball type and sum all balls across teams.

---

**Type 2: Partial Steps**

Step 1: We are to find the percentage of all the balls distributed that are Type A yellow balls. So, we need: (Total number of Type A yellow balls) / (Total number of all balls distributed) $\times$ 100%

Step 2: Calculate total number of Type A yellow balls. Team 1 has 72 members, and each player gets 4 Type A yellow balls. So Type A yellow balls = 72 $\times$ 4 = 288.

---

**Type 3: Explanation**

**Step 1: Understand what is being asked.** We are to calculate the percentage of all the balls distributed that are Type A yellow balls.

**Step 2: Calculate the number of Type A yellow balls.** Team 1 has 72 members. Each player gets 4 Type A yellow balls. Thus, Type A yellow balls $= 72 \times 4 = 288$.

**Step 3: Calculate total number of balls distributed.** We need to compute the total across all teams and all ball types.

Team 1 (Type A balls): $72 \times (4 + 3 + 3) = 72 \times 10 = 720$. Team 2 (Type B balls): $64 \times (5 + 4 + 2) = 64 \times 11 = 704$. Team 3 (Type C balls): $53 \times (3 + 2 + 6) = 53 \times 11 = 583$.

**Step 4: Add up total balls across all teams.** Total balls $= 720 + 704 + 583 = 2007$.

**Step 5: Calculate percentage of Type A yellow balls.** We already have Type A yellow balls = 288. So, Percentage = $\left(\frac{288}{2007}\right) \times 100$.

Simplify: $\frac{288}{2007} = \frac{96}{669} = \frac{32}{223}$. Now, $\frac{32}{223} \approx 0.1435$, and $0.1435 \times 100 = 14.35\%$.

**Final Answer:** $\boxed{14.35\%}$

**Why is this correct?**

- Only Team 1 gets Type A yellow balls.
- Total Type A yellow balls = $72 \times 4 = 288$.
- Total balls distributed = $720 + 704 + 583 = 2007$.
- Therefore, $(288/2007) \times 100 \approx 14.35\%$.

Thus, **14.35% of all distributed balls are Type A yellow balls.**

---

**Type 4: Ground Truth Answer**

I was told the answer is $\boxed{14.35\%}$. Do not blindly accept it. Take it as a reference and provide your own step-by-step reasoning.

---

### E.1.1 EXAMPLES OF SELF- AND GPT-O4-MINI-GENERATED ABSTRACT CUES

---

**Example 1**

**Question:** What is the maximum amount of heat that can be supplied to the system during the electrolysis of water without violating the second law of thermodynamics? (A) 237 kJ (B) 286 kJ (C) 163 kJ (D) 49 kJ (Correct answer: D)

**Hint (Self-generated):** Think about the difference between the energy required to break water (enthalpy) and the useful energy available (Gibbs free energy). This difference represents the minimum amount of energy that must be lost as heat due to entropy—this is a direct consequence of the second law of thermodynamics. What is this difference for water electrolysis?

**Hint (GPT-o4-mini-generated):** Use the relation $\Delta G = \Delta H - T\Delta S$ to find the entropy change, then recall that under a reversible (second-law-limit) process the maximum heat input is $Q_{max} = T\Delta S$.

---

**Example 2**

**Question:** When two fair dice are thrown, the numbers obtained are respectively $a$ and $b$. Determine the probability that the line $bx + ay = 1$ intersects the circle $x^2 + y^2 = 1$ and that the length of the intercepted chord does not exceed $\frac{4\sqrt{2}}{3}$ (Correct answer: $\frac{1}{9}$).

**Hint (Self-generated):** Focus on the geometric condition for a line to cut a chord of a given length in a unit circle. Recall that the length of the chord depends on the distance from the center of the circle to the line — use this relationship to set up an inequality involving $a$ and $b$. Then translate the condition into a constraint on $a^2 + b^2$.

**Hint (GPT-o4-mini-generated):** Use the fact that for the circle $x^2 + y^2 = 1$ and the line $bx + ay = 1$, the distance from the origin to the line is $d = \frac{1}{\sqrt{a^2+b^2}}$ and the chord-length is $L = 2\sqrt{1 - d^2}$ $(L \leq \frac{4\sqrt{2}}{3})$ to get a condition on $a^2 + b^2$, then count the integer pairs.

Table 6: Evaluation results on harder benchmarks. Similar to Table 1, NuRL shows superior performance compared to GRPO across models and benchmarks.

| | AIME | Brumo | CMIMC | HMMT (Feb / Nov) | OlymMATH (Easy / Hard) | Avg. |
|---|---|---|---|---|---|---|
| **Llama3.2-3B-Instruct** | | | | | | |
| Zero-shot | 0.00 | 3.33 | 1.72 | 0.83 / 3.33 | 2.50 / 2.50 | 2.03 |
| GRPO | **0.42** | **8.13** | 2.34 | 2.08 / 6.04 | 6.69 / 2.38 | 4.01 |
| NuRL (Self) | 0.21 | **8.13** | **2.81** | **2.50 / 7.92** | **7.19 / 2.69** | **4.49** |
| NuRL (GPT-o4-mini) | 1.67 | 7.08 | 2.03 | 0.83 / 7.08 | 6.94 / 2.38 | 4.00 |
| **OctoThinker-3B-Hybrid-Zero** | | | | | | |
| Zero-shot | 1.25 | 10.63 | 2.97 | 0.83 / 6.46 | 4.06 / 2.63 | 4.12 |
| GRPO | 4.58 | 11.25 | **3.44** | 2.08 / 6.46 | 5.62 / 2.12 | 5.08 |
| NuRL (Self) | **5.00** | **11.88** | 3.28 | **2.50 / 6.67** | **6.19 / 2.19** | **5.39** |
| NuRL (GPT-o4-mini) | 4.58 | 15.00 | 3.91 | 3.54 / 6.04 | 5.44 / 2.31 | 5.83 |
| **Qwen3-4B-Instruct-2507** | | | | | | |
| Zero-shot | 46.88 | 54.79 | 31.09 | 31.04 / 37.71 | 56.00 / 14.31 | 38.83 |
| GRPO | 50.42 | **61.67** | 32.97 | 33.33 / 42.08 | 56.12 / 15.62 | 41.74 |
| NuRL (Self) | **52.08** | 61.46 | **35.31** | **33.75 / 44.17** | **56.69 / 16.50** | **42.85** |
| NuRL (GPT-o4-mini) | 54.58 | 64.38 | 35.47 | 35.42 / 44.79 | 59.06 / 17.31 | 44.43 |

# F ADDITIONAL EVALUATIONS ON HARDER BENCHMARKS

In Table 6, we compare NuRL with GRPO on seven harder benchmarks from MathArena (Balunović et al., 2025) and OlymMATH (Sun et al., 2025). These datasets pose significant challenges for our models—for instance, Llama reaches only 2.03% zero-shot accuracy on average, and even Qwen achieves just 38.83%. Despite this difficulty, NuRL remains effective. On Qwen, for example, it attains 42.85%, representing gains of +4.02% over zero-shot and +1.11% over GRPO. Taken together, these results suggest that NuRL continues to offer reliable improvements, even on substantially harder reasoning benchmarks.

# G QUALITATIVE EXAMPLES OF GRPO AND NuRL OUTPUTS

We provide a qualitative example comparing GRPO and NuRL trained Qwen3-4B models. Notably, NuRL-trained models do not generate hint tokens at test time, so the format and style remain similar to GRPO models, despite the difference in performance stemming from NuRL being able to solve more training problems.

---

**Qualitative Example**

**Question:** Calculate the value of $\sqrt{9 + 8\cos 20°} - \sec 20°$.

**Qwen3-4B (GRPO) Output:**

To find the value of
$$\sqrt{9 + 8\cos 20°} - \sec 20°,$$
we will simplify the expression step by step.

First, let's simplify the expression inside the square root:
$$\sqrt{9 + 8\cos 20°}.$$

We can use the trigonometric identity for $\cos 20°$ and some algebraic manipulation to simplify this expression. However, let's first try to simplify the whole expression by using trigonometric identities and properties.

---

Notice that $\sec 20° = \frac{1}{\cos 20°}$.

So, the expression becomes

$$\sqrt{9 + 8\cos 20°} - \frac{1}{\cos 20°}.$$

Let's simplify

$$\sqrt{9 + 8\cos 20°}.$$

To proceed, we need to find a way to simplify $9 + 8\cos 20°$. However, without further trigonometric identities or specific values, we can consider using trigonometric properties and known values to simplify the expression.

Consider the expression

$$\sqrt{9 + 8\cos 20°} - \frac{1}{\cos 20°}.$$

We could attempt a common denominator or other identities, but a more direct approach is possible.

After recalculating and simplifying using trigonometric properties and correct identities, the expression becomes

$$\sqrt{9 + 8\cos 20°} - \sec 20°.$$

Using the proper simplification steps (omitted here for brevity), the final value is $\boxed{2}$.

---

**Qwen3-4B (NuRL) Output:**

We are asked to compute:

$$\sqrt{9 + 8\cos 20°} - \sec 20°$$

Let us denote:

$$x = \sqrt{9 + 8\cos 20°} - \sec 20°$$

We want to find the value of $x$.

**Step 1: Consider simplification**

We first try a numerical approximation to guess the value.

**Step 2: Numerical approximation**

Compute $\cos 20°$ and $\sec 20°$:

$$\cos 20° \approx 0.9397, \quad \sec 20° = \frac{1}{\cos 20°} \approx 1.0642$$

Now compute:

$$8\cos 20° \approx 7.5176, \quad 9 + 8\cos 20° \approx 16.5176$$

Then:

$$\sqrt{16.5176} \approx 4.064$$

Subtracting $\sec 20°$:

$$x \approx 4.064 - 1.0642 \approx 2.9998 \approx 3$$

Hence, we suspect:

$$\sqrt{9 + 8\cos 20°} - \sec 20° = 3$$

**Step 3: Algebraic verification**

We want to verify:

$$\sqrt{9 + 8\cos 20°} = 3 + \sec 20°$$

Square both sides:

$$9 + 8\cos 20° = (3 + \sec 20°)^2 = 9 + 6\sec 20° + \sec^2 20°$$

Subtract 9:

$$8\cos 20° = 6\sec 20° + \sec^2 20°$$

Let $c = \cos 20°$, so $\sec 20° = \frac{1}{c}$. Then:

$$8c = \frac{6}{c} + \frac{1}{c^2}$$

Multiply both sides by $c^2$:

$$8c^3 = 6c + 1 \quad \Rightarrow \quad 8c^3 - 6c - 1 = 0$$

Check with the triple-angle identity:

$$\cos 3\theta = 4\cos^3 \theta - 3\cos \theta$$

Set $\theta = 20°$, then $\cos 60° = \frac{1}{2}$:

$$4c^3 - 3c = \frac{1}{2} \quad \Rightarrow \quad 8c^3 - 6c - 1 = 0$$

This confirms the equation holds. Therefore:

$$\sqrt{9 + 8\cos 20°} = 3 + \sec 20°$$

So finally:

$$\sqrt{9 + 8\cos 20°} - \sec 20° = 3$$

Final Answer: $\boxed{3}$

