# OpenReview forum: "Nudging the Boundaries of LLM Reasoning"
_ICLR.cc/2026/Conference — ICLR 2026 Poster_

### Official Review · Reviewer_1SjR · 2025-10-28

**Soundness:** 3
**Presentation:** 4
**Contribution:** 3
**Rating:** 6
**Confidence:** 5

**Summary:**

This paper introduces NuRL, a method to leverage off-policy hints (either self-generated or generated by teacher) that are (i) abstractions (ii) partial solutions (iii) explanations and (iv) ground truth answers. These are done for difficult problems when accuracy is zero.

Interestingly, by injecting these hints during GRPO when rewards are all zero, the model manages to achieve superior performance during RL (do they show superior performance on specifically these hard problems? see my question in the Question section below).

Comparison is done against vanilla GRPO and there seems to be a 1-2% increase on several benchmarks.

**Strengths:**

- The problem setting is novel and interesting - getting GRPO to sample non-zero rewards on difficult problems
- The method is simple and effective
- very nice diagrams
- paper well-written, easy to understand

**Weaknesses:**

- not enough difficult benchmarks, esp. for Qwen3-4B-Instruct which is already at 70% performance even before RL
- I'd like to see some more qualitative rollout examples, mentioned in questions
- more comparison against other methods

**Questions:**

1. Qwen3-4B seems to achieve already such a high accuracy on the benchmarks. I believe it would strengthen your paper if you can evaluate on more difficult benchmarks. For the rebuttal, I'd like to see results on ALL benchmarks listed in the final answer competitions in the standard Math Arena: https://matharena.ai/?comp=overall--final-answer_competitions. In addition, please add evaluations on OlymMath-EN: https://github.com/RUCAIBox/OlymMATH/tree/main/data. If these are still too easy for Qwen3-4B base (>50% accuracy), you can also look into evaluating on Omni-MATH.
2. What's the difference between before hint and after hint in figure 6? Can I see another curve which evaluates the model's capability *without the hint*, after training on rollouts *conditioned on the hint*?
3. Related to the above question, why do you think training on rollouts conditioned on the hint should improve rollouts without the hint? If the hint tokens are not upweighted, then I don't understand why they would be sampled at test time. And if they are not sampled, the rollouts shouldn't improve?
4. Can you compare and contrast example rollouts on difficult problems (i) without the hint, (ii) given the hint, and (iii) after training on the hint-conditioned rollouts, without the hint?
5. In figure 4, what is the performance of the models without any hints? Could you add a bar next to them showing that
6. How do you compare against baselines where you SFT on the partial solutions? How about against other methods that learn directly on correct traces such as LUFFY (Gemini generated traces) and Critique-GRPO (self-revised correct races)?

If you can make this paper more sound and answer the questions above with solid results, I will raise my score and vouch for you.

---

> ### Author Response · Authors · 2025-11-24
> **Response to Reviewer 1SjR (Part 1)**
>
> We thank the reviewer for appreciating our work as “novel and interesting” and “simple and effective”. Please find the answers to your questions below.
>
> ### \[Q1\] Not Enough Difficult Benchmarks
>
> > I'd like to see results on ALL benchmarks listed in the final answer competitions in the standard Math Arena
>
> We thank the reviewer for the insightful feedback, and we fully agree that evaluating on even harder benchmarks would help consolidate the contribution. Following your suggestion, **we evaluate all three models on 7 additional benchmarks.**
>
> For MathArena, we evaluated AIME 2025, Brumo 2025, CMIMC 2025, HMMT 2025 (Feb) and HMMT 2025 (Nov). We didn’t find a released test set for SMT 2025, but we include the latest Nov HMMT. For OlymMATH, we evaluated the English subset for both easy and hard. Results are also averaged over 16 runs, the same as our evaluation protocol used throughout the paper.
>
> &nbsp;
>
> **Llama3.2 3B**
>
> ---
>
> | Method               | AIME 2025 | Brumo 2025 | CMIMC 2025 | HMMT 2025 (Feb) | HMMT 2025 (Nov) | OlymMATH Easy | OlymMATH Hard | Avg. |
> |----------------------|-----------|------------|------------|------------------|------------------|----------------|----------------|------|
> | Zero-shot            | 0.00      | 3.33       | 1.72       | 0.83             | 3.33             | 2.50           | 2.50           | 2.03 |
> | GRPO                 | **0.42**      | **8.13**       | 2.34       | 2.08             | 6.04             | 6.69           | 2.38           | 4.01 |
> | NuRL (Self)          | 0.21      | **8.13**       | **2.81**       | **2.50**             | **7.92**             | **7.19**           | **2.69**           | **4.49** |
> | NuRL (GPT-o4-mini)   | 1.67      | 7.08       | 2.03       | 0.83             | 7.08             | 6.94           | 2.38           | 4.00 |
>
> &nbsp;
> &nbsp;
>
> **OctoThinker 3B**
>
> ---
>
> | Method               | AIME 2025 | Brumo 2025 | CMIMC 2025 | HMMT 2025 (Feb) | HMMT 2025 (Nov) | OlymMATH Easy | OlymMATH Hard | Avg. |
> |----------------------|-----------|------------|------------|------------------|------------------|----------------|----------------|------|
> | Zero-shot            | 1.25      | 10.63      | 2.97       | 0.83             | 6.46             | 4.06           | 2.63           | 4.12 |
> | GRPO                 | 4.58      | 11.25      | **3.44**       | 2.08             | 6.46             | 5.62           | 2.12           | 5.08 |
> | NuRL (Self)          | **5.00**      | **11.88**      | 3.28       | **2.50**             | **6.67**             | **6.19**           | **2.19**           | **5.39** |
> | NuRL (GPT-o4-mini)   | 4.58      | 15.00      | 3.91       | 3.54             | 6.04             | 5.44           | 2.31           | 5.83 |
>
> &nbsp;
> &nbsp;
>
> **Qwen3 4B**
>
> ---
>
> | Method               | AIME 2025 | Brumo 2025 | CMIMC 2025 | HMMT 2025 (Feb) | HMMT 2025 (Nov) | OlymMATH Easy | OlymMATH Hard | Avg. |
> |----------------------|-----------|------------|------------|------------------|------------------|----------------|----------------|------|
> | Zero-shot            | 46.88     | 54.79      | 31.09      | 31.04            | 37.71            | 56.00          | 14.31          | 38.83 |
> | GRPO                 | 50.42     | **61.67**      | 32.97      | 33.33            | 42.08            | 56.12          | 15.62          | 41.74 |
> | NuRL (Self)          | **52.08**     | 61.46      | **35.31**      | **33.75**            | **44.17**            | **56.69**          | **16.50**          | **42.85** |
> | NuRL (GPT-o4-mini)   | 54.58     | 64.38      | 35.47      | 35.42            | 44.79            | 59.06          | 17.31          | 44.43 |
>
> The overall trend aligns with what we observed in Table 1, even though these datasets are considerably more challenging. Specifically, GRPO consistently improves over the base model, and NuRL provides additional gains on top of the improvements achieved by GRPO.

---

> ### Author Response · Authors · 2025-11-24
> **Response to Reviewer 1SjR (Part 2)**
>
> ### \[Q2\] What's the difference between before hint and after hint in figure 6?
>
> We appreciate the reviewer’s question. In our pipeline, hints are appended only when all rollouts initially fail. Thus, for each training query we (a) first generate G rollouts without hints, and if none solve the problem, we (b) append the hint and generate another G rollouts with hints. Figure 6 reports the solvable fraction under these two conditions—before hint corresponds to (a), and after hint corresponds to (b).
>
> > Can I see another curve which evaluates the model's capability without the hint, after training on rollouts conditioned on the hint?
>
> This is in fact how we evaluate by default: training uses hints, but evaluation never uses hints. To check for any train–test mismatch, we additionally evaluated a variant with test-time hints. Because gold answers are unavailable at evaluation, we use an 8-shot prompt of example abstract hints and let the model generate its own hints for each test sample.
>
> | Method      | MATH 500 | MATH Hard | AIME | GPQA  | MMLU-Pro | Date  | Avg.  |
> |------------|----------|-----------|------|-------|----------|-------|-------|
> | NuRL  w test-time hints     | 56.44    | 29.01     | 8.33 | 27.19| 35.48    | 60.20 |36.11 |
> | NuRL  w/o test-time hints     | **58.04**    | **31.62**     | **9.17** | **28.28** | **36.18**    | **61.65** | **37.49** |
>
> Providing hints at test time slightly decreases performance, suggesting that the model does not overfit to hints and that any distribution mismatch is mild.
>
> Therefore, both Table 1 and Figure 6 already reflect the model’s true capability without hints, even though training uses hint-conditioned rollouts.
>
> &nbsp;
> &nbsp;
>
> ### \[Q3\] Why Should Hint-conditioned Training Improve Rollouts without the Hint?
>
> > If the hint tokens are not upweighted, then I don't understand why they would be sampled at test time. And if they are not sampled, the rollouts shouldn't improve?
>
> We thank the reviewer for the insightful question. This is highly connected to Q2: we do not expect the model to generate hint-like text at test time, nor do we upweight hint tokens. This setup is intentional: the hints serve as an auxiliary training signal that shapes the model’s internal representations, but the model is not expected to generate or rely on these tokens at inference.
>
> Importantly, **gradients do not backpropagate into the user-prompt tokens** (where our hint tokens are). The RL loss only updates parameters based on model-generated tokens, so the model does not learn to depend on the presence of hint tokens in the input. Instead, the hints influence optimization indirectly, by providing a light-weight guidance that improves credit assignment. With hints present, the model is nudged toward more promising reasoning trajectories, reducing the sparsity of the RL signal and enabling more efficient policy improvement.
>
> Empirically, we observe that models trained with our method do not generate hint tokens or exhibit format drift at test time. Their outputs closely match the GRPO model’s style, indicating that the model internalizes better reasoning patterns rather than relying on the hints themselves. **This qualitative result is provided in Appendix G in the revised PDF.**
>
> Additionally, Figure 6 shows a steady increase in the solvable-problem fraction during training. Because hints are provided only when a problem is initially unsolvable, this rise from 66% to 70% indicates that the model **increasingly succeeds without needing hints** at all. This trend suggests that the model is not relying on hints, but instead learning to solve problems independently.
>
> &nbsp;
> &nbsp;
>
> ### \[Q4\] Qualitative Examples
>
> > Can you compare and contrast example rollouts on difficult problems (i) without the hint, (ii) given the hint, and (iii) after training on the hint-conditioned rollouts, without the hint?
>
> Based on the above responses that clarified we did not use hints at test time by default, we believe the best way to compare qualitative examples is to compare the output generated by GRPO and the output generated by our NuRL method. **To better visualize the outputs in LaTeX, we provide the example in Appendix G (lines 1003-1129)** in the revised PDF. Since our trained models do not generate hint tokens at test time, the format and style remain similar to GRPO models, despite the difference in performance stemming from our model being able to solve more training problems.
>
> &nbsp;
> &nbsp;
>
> ### \[Q5\] Hint Type Comparison
>
> > In figure 4, what is the performance of the models without any hints? Could you add a bar next to them showing that
>
> Based on the clarification above, we note that Figure 4 shows model performance without any hints at test time. The comparison of hint types is made by varying the hints used during training, while all models are evaluated without hints. We hope this also helps clarify the point\!

---

> ### Author Response · Authors · 2025-11-24
> **Response to Reviewer 1SjR (Part 3)**
>
> ### \[Q6\] More Baselines
>
> > How do you compare against baselines where you SFT on the partial solutions?
>
> In our SFT baselines, RFT and ReFT, we did not train on the partial solutions, but instead, let the model first generate 8 rollouts, filter out the incorrect ones, and then SFT on the remaining correct self-generated chain-of-thought. When comparing with NuRL using self-generated hints, this is comparable because none of the methods rely on external models.
>
> &nbsp;
> &nbsp;
>
> > How about against other methods that learn directly on correct traces such as LUFFY (Gemini generated traces) and Critique-GRPO (self-revised correct races)?
>
> We thank the reviewer for this insightful question. We agree that including comparisons with Critique-GRPO and LUFFY strengthens our paper. We conducted additional experiments on LLaMA3.2 3B for this purpose.
>
> | Method      | MATH 500 | MATH Hard | AIME | GPQA  | MMLU-Pro | Date  | Avg.  |
> |------------|----------|-----------|------|-------|----------|-------|-------|
> | Zero-shot  | 35.71    | 15.28     | 3.33 | 13.23 | 11.52    | 2.33  | 13.57 |
> | NuRL (Self)       | **58.04**    | **31.62**     | **9.17** | **28.28** | **36.18**    | 61.65 | **37.49** |
> | Critique-GRPO | 56.92 | 30.37 | 8.33 | 27.32 | 34.46 | **62.77** | 36.69 |
>
> We observe that NuRL with self-generated hints slightly outperforms Critique-GRPO by 0.8% on average, suggesting that providing hints may be slightly more effective than simply informing the model that a previous answer is incorrect and letting it self-refine. We do think these approaches are not mutually exclusive and could potentially be complementary.
>
> Next, we compared with LUFFY, using GPTo4-mini to generate the full CoT trajectories for injection into rollouts. The key difference is that LUFFY leverages complete reasoning chains from a stronger model, while NuRL uses high-level hints without access to full trajectories.
>
> | Method      | MATH 500 | MATH Hard | AIME | GPQA  | MMLU-Pro | Date  | Avg.  |
> |------------|----------|-----------|------|-------|----------|-------|-------|
> | NuRL (GPTo4-mini) | 59.30 | 32.42 | 10.83 | 28.40 | 41.38 | 63.53 | 39.31 |
> | LUFFY (GPTo4-mini) | 60.29 | 33.03 | 11.33 | 28.32 | 42.52 | 63.02 | 39.75 |
>
> We find that LUFFY performs comparably to NuRL when provided with GPT-generated hints, indicating that high-level hints capture most of the benefits of full reasoning trajectories from a stronger model.
>
> **We have included all baselines, benchmarks, and new results in Appendix F (lines 972-1001 highlighted in blue)**. We thank the reviewer for their helpful feedback, which allowed us to strengthen our contribution, and we hope this addresses all of your questions.

---

### Official Review · Reviewer_RKzS · 2025-10-31

**Soundness:** 3
**Presentation:** 3
**Contribution:** 3
**Rating:** 6
**Confidence:** 4

**Summary:**

This paper proposes NuRL, a reinforcement learning method designed to push the reasoning capability limits of LLMs by enabling learning from tasks that are currently unsolvable under the base policy. Built on GRPO, NuRL incorporates self-generated abstract hints that are injected only when all rollouts fail, turning zero-reward samples into informative training signals. Through experiments on six reasoning benchmarks (MATH 500, MATH Hard, AIME 2024, GPQA, MMLU-Pro, Date Understanding) and three 3–4B models (Llama, OctoThinker, Qwen), the authors demonstrate consistent improvements in pass@1 and pass@k.

**Strengths:**

- The paper tackles a timely and important limitation of reasoning-oriented RL methods.
- The experimental evaluation is comprehensive, spanning diverse tasks and multiple model scales.
- The analysis is thorough, including insightful studies on hint type, hint timing, pass@k, and solvable-problem curves.
    - I really like Figure 4. It clearly demonstrates the relative effectiveness of different hint types.
- The proposed two-stage selective-hint strategy is novel, well-motivated, and addresses the problem effectively.

**Weaknesses:**

- The evaluation is limited to models up to 4B parameters. As model capacity increases, the proportion of unsolvable problems naturally decreases. It is unclear whether the method would still be effective when applied to larger, more capable models.
- There is a clear train and test distribution shift where hints are provided during training but absent during evaluation.
- The Self-Consistency method lacks sufficient detail. Although it is not the focus of the paper, additional implementation details would provide clearer context.

**Questions:**

Please refer to the weaknesses.

---

> ### Author Response · Authors · 2025-11-24
> **Response to Reviewer RKzS (Part 1)**
>
> We thank the reviewer for recognizing our work “tackles a timely and important limitation of reasoning-oriented RL methods”, “evaluation is comprehensive”, “analysis is thorough”, and “the proposed two-stage selective-hint strategy is novel, well-motivated, and addresses the problem effectively”. We are also glad to see that the reviewer likes Figure 4\. Please find the answers to your questions below.
>
> &nbsp;
> &nbsp;
>
> ### \[W1\] Evaluation Beyond 4B-Parameter Models
>
> > The evaluation is limited to models up to 4B parameters.
>
> We thank the reviewer for raising this point. While our experiments primarily focus on 3–4B models due to computational constraints—GRPO-based methods require 6–7 days to converge on a 4B model, and scaling to 7–8B models would demand significantly longer training time—we note that our study includes three distinct model families. This diversity suggests that the observed performance trends are not specific to a single architecture and are likely to generalize across larger models as well.
>
> &nbsp;
> &nbsp;
>
> > As model capacity increases, the proportion of unsolvable problems naturally decreases
>
> We appreciate this insightful observation and fully agree. In practice, one can assess the distribution of problem difficulty before training begins. For example, by generating 8 samples per query, it is straightforward to estimate the proportion of problems with a 0/8 pass rate. Because this pass-rate distribution is model-dependent, different model sizes or families will yield different proportions of “hard” problems. When operating with sufficiently large datasets, it is feasible to identify and curate a substantial subset of genuinely challenging examples—even for larger and stronger models.
>
> &nbsp;
> &nbsp;
>
> ### \[W2\] Train-Test Distribution Shift
>
> > There is a clear train and test distribution shift where hints are provided during training but absent during evaluation
>
> We thank the reviewer for this insightful observation. We agree that there is a train-test mismatch, since hint tokens appear during training but not at test time. This setup is intentional: the hints serve as an auxiliary training signal that shapes the model’s internal representations, but the model is not expected to generate or rely on these tokens at inference.
>
> Importantly, **gradients do not backpropagate into the user-prompt tokens** (where our hint tokens are). The RL loss only updates parameters based on model-generated tokens, so the model does not learn to depend on the presence of hint tokens in the input. Instead, the hints influence optimization indirectly, by providing a light-weight guidance that improves credit assignment. With hints present, the model is nudged toward more promising reasoning trajectories, reducing the sparsity of the RL signal and enabling more efficient policy improvement.
>
> Empirically, we observe that models trained with our method do not generate hint tokens or exhibit format drift at test time. Their outputs closely match the GRPO model’s style, indicating that the model internalizes better reasoning patterns rather than relying on the hints themselves. **This qualitative result is provided in Appendix G in the revised PDF.**
>
> Additionally, Figure 6 shows a steady increase in the solvable-problem fraction during training. Because hints are provided only when a problem is initially unsolvable, this rise from 66% to 70% indicates that the model **increasingly succeeds without needing hints** at all. This trend suggests that the model is not relying on hints, but instead learning to solve problems independently, further mitigating concerns about the train–test mismatch.
>
> We also conducted an additional experiment by providing hints at test time to investigate this mismatch more deeply. Since gold answers are unavailable during evaluation, we employ an 8-shot prompt showcasing examples of good abstract hints, and ask the model to generate analogous hints for each test sample. We then evaluate NuRL under these conditions.
>
> | Method      | MATH 500 | MATH Hard | AIME | GPQA  | MMLU-Pro | Date  | Avg.  |
> |------------|----------|-----------|------|-------|----------|-------|-------|
> | NuRL  w test-time hints     | 56.44    | 29.01     | 8.33 | 27.19| 35.48    | 60.20 |36.11 |
> | NuRL  w/o test-time hints     | **58.04**    | **31.62**     | **9.17** | **28.28** | **36.18**    | **61.65** | **37.49** |
>
> Overall, these analyses and additional experiments demonstrate that the train–test mismatch posed by hint tokens does not undermine our method’s effectiveness. Hints serve purely as an auxiliary training signal, shaping internal reasoning without creating dependence at inference. The model consistently solves more problems over training, maintains output style, and performs slightly better without test-time hints, confirming that **NuRL internalizes improved reasoning rather than relying on hints.**

---

> ### Author Response · Authors · 2025-11-24
> **Response to Reviewer RKzS (Part 2)**
>
> ### \[W3\] Details about Self-Consistency
>
> > The Self-Consistency method lacks sufficient detail
>
> We thank the reviewer for this question and the opportunity to clarify our setup. The Self-Consistency method used in Figure 3 follows the standard procedure introduced in the original paper: for each test sample, we sample 16 independent CoT outputs, extract the final answer from each, and then select the majority answer as the final prediction. All generation hyperparameters are identical to those used in our other evaluation settings (detailed in lines 305–321).
>
> For example, the following 16 CoT are generated for a test sample:
>
> | CoT Sample | Predicted Answer |
> | ---------- | ---------------- |
> | CoT 1      | (A)              |
> | CoT 2      | (B)              |
> | CoT 3      | (B)              |
> | CoT 4      | (D)              |
> | CoT 5      | (B)              |
> | …          | …                |
> | CoT 16     | (B)              |
>
> In this case, the final predicted answer is (B) as it is the majority answer.
>
> To make the analysis more complete, we have also added Qwen’s performance—with and without Self-Consistency—to Figure 3 in the revised PDF. **These updates are noted in lines 359–371 (highlighted in blue).**

---

> ### Comment · Reviewer_RKzS · 2025-11-24
>
> I thank the authors for the detailed feedback! I will maintain the current score.

---

### Official Review · Reviewer_GxNS · 2025-11-01

**Soundness:** 2
**Presentation:** 3
**Contribution:** 2
**Rating:** 4
**Confidence:** 4

**Summary:**

The paper proposes NuRL (Nudging LLMs with Reinforcement Learning), a method designed to extend the reasoning boundaries of large language models (LLMs) by enabling learning from previously unsolvable problems. Standard online RL methods like GRPO fail on such problems because all rollouts receive zero rewards, preventing gradient updates. NuRL introduces self-generated hints to “nudge” the model and make these problems solvable. During training, hints are injected only when all rollouts for a problem fail, and only for some rollouts to avoid overfitting.

**Strengths:**

1. The studied problem is important, and the proposed idea is interesting: knowledge outside of rl training can help the model further improve the performance.

2. The evaluation shows that the proposed method achieves consistent performance improvement compared to GRPO.

3.  Ablation studies carefully examine what types of hints help (abstract vs. explicit), when to apply them.

**Weaknesses:**

1. One of my concerns is where the accuracy improvement comes from. As the authors mentioned: “In practice, such uninformative problems (both too hard and too easy) are often discarded to improve training efficiency”. However, besides discarding those examples, a common practice (e.g., DAPO), will resample and fill the batch (dynamic sampling). Simply discarding the example can hurt the training performance. That’s also why dynamic sampling can improve the model performance and stabilize training. A side effort of NuRL is to address this zero advantage issue, so the performance improvement may come from more stable optimization rather than hints. It would be great to compare NuRL and dynamic sampling to see how those hints can improve the performance.

2. Besides hints from the base model, the paper also proposes to use another more powerful model (e.g., GPT-4o) to generate hints. Such an offline method has already been studied in [1]. It would be great to include it as a baseline for comparison.


[1] Yan, Jianhao, Yafu Li, Zican Hu, Zhi Wang, Ganqu Cui, Xiaoye Qu, Yu Cheng, and Yue Zhang. "Learning to reason under off-policy guidance." arXiv preprint arXiv:2504.14945 (2025).

**Questions:**

listed in weakness

---

> ### Author Response · Authors · 2025-11-24
> **Response to Reviewer GxNS**
>
> We thank the reviewer for acknowledging that “the studied problem is important”, “the proposed idea is interesting”, and “the proposed method achieves consistent performance improvement”. Please find the answers to your questions below.
>
> &nbsp;
> &nbsp;
>
> ### \[W1\] Where the accuracy improvement comes from
>
> We thank the reviewer for raising this point. To clarify, we ensure that the GRPO baseline we compare against is as directly comparable as possible; the primary difference is whether hints are used or not. As shown in Table 1, incorporating hints consistently improves performance across three models and six datasets.
>
> &nbsp;
> &nbsp;
>
> > Simply discarding the example can hurt the training performance.
>
> We understand that part of the improvement could stem from stabilizing training. However, we emphasize that one of our contributions is making “unsolvable problems learnable.” In contrast, dynamic sampling in DAPO addresses this from a different perspective: it ensures that each batch contains samples with neither all-zero nor all-one rewards, avoiding “too easy” or “too hard” examples in the training data. In this setting, NuRL would not apply, since hints are only injected when all rewards are zero. Therefore, NuRL and DAPO with dynamic sampling are less directly comparable, as the training data distributions differ.
>
> &nbsp;
> &nbsp;
>
> > It would be great to compare NuRL and dynamic sampling to see how those hints can improve the performance.
>
> Following your suggestion, we conducted an additional experiment using DAPO’s dynamic sampling strategy with the same training samples we used in NuRL using Llama3.2 3B.
>
> | Method      | MATH 500 | MATH Hard | AIME | GPQA  | MMLU-Pro | Date  | Avg.  |
> |------------|----------|-----------|------|-------|----------|-------|-------|
> | Zero-shot  | 35.71    | 15.28     | 3.33 | 13.23 | 11.52    | 2.33  | 13.57 |
> | DAPO       | 56.02    | 28.71     | 8.33 | **28.98** | 34.03    | 56.20 | 35.38 |
> | NuRL       | **58.04**    | **31.62**     | **9.17** | 28.28 | **36.18**    | **61.65** | **37.49** |
>
> On average, NuRL outperforms dynamic sampling GRPO by 2.11%, indicating that the performance gains primarily stem from the incorporation of hints.
>
> &nbsp;
> &nbsp;
>
> ### \[W2\] Comparison to an offline baseline
>
> > Besides hints from the base model, the paper also proposes to use another more powerful model (e.g., GPT-4o) to generate hints. Such an offline method has already been studied in \[1\]. It would be great to include it as a baseline for comparison.
>
> We thank the reviewer for this suggestion. We would like to clarify that using GPT-generated hints is **not the primary focus** of our work; we include it mainly to provide a comprehensive analysis that explores variations in hint type and source.
>
> Nonetheless, we agree that LUFFY is a meaningful baseline for reference. While we already included this work in our original submission, we now present new experimental results on Llama below:
>
> | Method      | MATH 500 | MATH Hard | AIME | GPQA  | MMLU-Pro | Date  | Avg.  |
> |------------|----------|-----------|------|-------|----------|-------|-------|
> | Zero-shot  | 35.71    | 15.28     | 3.33 | 13.23 | 11.52    | 2.33  | 13.57 |
> | NuRL (Self)       | 58.04    | 31.62     | 9.17 | 28.28 | 36.18    | 61.65 | 37.49 |
> | NuRL (GPTo4-mini) | 59.30 | 32.42 | 10.83 | 28.40 | 41.38 | 63.53 | 39.31 |
> | LUFFY (GPTo4-mini) | 60.29 | 33.03 | 11.33 | 28.32 | 42.52 | 63.02 | 39.75 |
>
> The main difference between LUFFY and NuRL is that **LUFFY leverages the full reasoning chain generated by GPT-o4-mini**, whereas NuRL uses only **high-level hints** without access to the complete reasoning trajectories from the stronger model. We observe that LUFFY performs comparably to NuRL when provided with GPT-generated hints, demonstrating that high-level hints can capture most of the benefits of full reasoning trajectories.

---

### Official Review · Reviewer_1qwq · 2025-11-02

**Soundness:** 3
**Presentation:** 3
**Contribution:** 3
**Rating:** 4
**Confidence:** 3

**Summary:**

The submission introduces NuRL, which augments typical LLM RL training by providing hints when all responses to a query are incorrect. Experiments consistently show that this method can address the problem of zero gradients for such questions, allowing the model to solve more problems. Ablation studies illustrate helpful properties of hints and support implementation choices.

**Strengths:**

**Originality**  The hinting implementation's details are ablated and investigated, clarifying how to provide helpful hints and providing potential explanations for why specific hinting schemes work. The method is evaluated on a diverse range of reasoning problems using models with different RL behavior profiles, which is uncommon in RL studies and adds rigor.

**Quality** Experiments are well designed and rigorously test questions of interest. For example, the selection of models tested is well motivated (line 246).

**Clarity** The paper is well written. Figures and tables clearly communicate the method and experimental results. Background work is appropriately referenced.

**Significance** The method improves pass@1 and pass@1024 performance on key RL tasks, suggesting an improved ability to solve hard problems, which the community will be interested in. Ablation studies that characterize properties of good hints can guide future work in this area.

**Weaknesses:**

There are some highly related works, namely Huang et al. (2025), which the authors cite. The submission is still novel, however, as it focuses on LLMs instead of MLLMs, self-generated hints instead of relying on an external model or dataset, and investigation of the properties of good hinting schemes.

**Questions:**

I would be happy to improve my rating after some minor changes that reinforce the paper's existing strengths, which are its exploration of various hint implementations and careful experimental designs.

- Line 136: For RL on LLMs with experience replay buffers, in addition to Lu et al. (2025), I would cite Bartoldson et al. (2025), “Trajectory Balance with Asynchrony”.

- Line 205: How is the amount of information measured in order to support this statement about the ordering?

- Line 313: Could you please explain this procedure more clearly? My understanding is that a model is trained with GRPO until performance plateaus, then it's split into two copies: one that NuRL training is applied to, and another that continued GRPO training is applied to. If this is correct, would it make sense to create another GRPO baseline that -- at the halfway point -- also switches to the 8 responses per query that NuRL uses?

- Line 363: “complementary”

- Figure 3: why exclude Qwen from this experiment?

- Figure 6: it looks like NuRL performance would continue to improve with longer training. For example, if you train for 1000 steps, do you see a plateau with NuRL?

- Figure 6: it would be interesting to augment this study to test if more direct hints improve the solvable fraction on train data but not test data.

---

> ### Author Response · Authors · 2025-11-24
> **Response to Reviewer 1qwq (Part 1)**
>
> We sincerely thank the reviewer for recognizing our work’s originality (“details are ablated and investigated”), quality (“experiments are well designed and rigorously test questions of interest”), clarity (“The paper is well written”) and significance (“improves pass@1 and pass@1024 performance on key RL tasks”). Please find the answers to your questions below.
>
> ---
> &nbsp;
>
> > \[W1\] There are some highly related works, namely Huang et al. (2025), which the authors cite. The submission is still novel…
>
> We are grateful that the reviewer recognizes our focus on language models, self-generated hints, and the examination of effective hinting schemes as meaningful contributions. While our work shares conceptual similarities with prior work, such as the general idea of providing hints, we approach a distinct research question: *how can a model broaden its own knowledge boundaries **without relying on an external model?***
>
> Huang et al. \[1\] build on the Llava-CoT \[2\] dataset, which follows a four-stage reasoning structure (Summary, Caption, Reasoning, Conclusion). As a result, each Llava-CoT sample is produced through an iterative prompting pipeline that **requires four separate GPT-4o calls.** In contrast, our work centers on **self-generated hints.**
>
> As the reviewer kindly noted, we conduct an extensive investigation into different types and sources of hints (Figure 4), and we find that high-level, self-generated abstract cues are particularly effective in pushing the model’s boundaries.
>
> We thank the reviewer again for their generous assessment and helpful framing of our contributions.
>
> * \[1\] https://arxiv.org/abs/2503.23905
> * \[2\] https://arxiv.org/abs/2411.10440
>
> &nbsp;
> &nbsp;
>
> > \[Q1\] Line 136: For RL on LLMs with experience replay buffers, in addition to Lu et al. (2025), I would cite Bartoldson et al. (2025), “Trajectory Balance with Asynchrony”.
>
> We thank the reviewer for the pointer. **We agree that it is relevant, and we have included this work in the related work (highlighted in blue in the revised PDF, lines 137-140).**
>
> &nbsp;
> &nbsp;
>
> > \[Q2\] Line 205: How is the amount of information measured in order to support this statement about the ordering?
>
> This ordering is based on how each hint type is generated and how many layers of processing separate it from the gold answer:
>
> * **Abstract cues**: the most abstract hints, produced by abstracting an explanation of why the answer is correct into generalized guidance.
> * **Partial correct steps**: one level closer to the gold answer; generated by producing a step-by-step solution conditioned on the correct answer, but without applying any abstraction.
> * **Explanations**: answer-conditioned explanations that justify why the answer is correct. These sometimes contain phrases like “but the answer is X, so I should try something else…,” which can inadvertently leak the answer and therefore sit even closer to the gold answer in terms of information content.
> * **Ground-truth answer**: the most explicit and direct form of answer information.
>
> We refer the reviewer to Appendix E.1 for concrete examples of each hint type.
>
> &nbsp;
> &nbsp;
>
> > \[Q3\] Line 313: Could you please explain this procedure more clearly? My understanding is that a model is trained with GRPO until performance plateaus, then it's split into two copies: one that NuRL training is applied to, and another that continued GRPO training is applied to. If this is correct, would it make sense to create another GRPO baseline that \-- at the halfway point \-- also switches to the 8 responses per query that NuRL uses?
>
> Your understanding is correct\! The only small clarification is that the “continued GRPO copy” keeps using 16 rollouts throughout training. This is actually a stronger setting than the 8-rollout setup used by NuRL, so the GRPO baseline is given more opportunities per query. Even under this favorable configuration, NuRL still outperforms it, which we highlighted in lines 333-337: “Importantly, these improvements are achieved on top of already strong GRPO performance, **despite GRPO using 16 rollouts per question, whereas NuRL uses only 8** (with the option to generate another eight only if all initial rollouts fail)”.
>
> &nbsp;
> &nbsp;
>
> > \[Q4\] Line 363: “complementary”
>
> Thanks for catching the typo\! We fixed this in the revised PDF (highlighted in blue).

---

> ### Author Response · Authors · 2025-11-24
> **Response to Reviewer 1qwq (Part 2)**
>
> > \[Q5\] Figure 3: why exclude Qwen from this experiment?
>
> We thank the reviewer for the careful review and insightful feedback. Qwen was initially excluded for better clarity and visualization, as the overall trend is similar across models. To provide a complete view, **we now include a comprehensive table with Qwen** and have **updated Figure 3 and lines 370–371 in the revised PDF** accordingly (highlighted in blue).
>
> | Model / Method        | Zero-shot | GRPO   | NuRL (Self) | NuRL (o4-mini) |
> |----------------------|-----------|--------|-------------|----------------|
> | Llama3.2-3B (Avg@16) | 35.71     | 56.76  | 58.04       | 59.30          |
> | Llama3.2-3B (SC@16)  | 55.60      | 64.40 (+7.64)  | 66.00   **(+7.96)**    | 67.60           |
> | OctoThinker-3B (Avg@16) | 59.98  | 65.80  | 67.40        | 69.21          |
> | OctoThinker-3B (SC@16)  | 66.20   | 73.60 (+7.80)   | 76.80   **(+9.40)**     | 78.00           |
> | Qwen3-4B (Avg@16)    | 94.88     | 96.52  | 96.46       | 96.58          |
> | Qwen3-4B (SC@16)     | 96.60     | 97.60  (+1.08) | 97.62   **(+1.16)**     | 97.40           |
>
> Overall, the trend remains consistent: across all three models, those trained with NuRL gain more from applying Self-Consistency than their GRPO-only counterparts.
>
> &nbsp;
> &nbsp;
>
> > \[Q6\] Figure 6: it looks like NuRL performance would continue to improve with longer training. For example, if you train for 1000 steps, do you see a plateau with NuRL?
>
> We thank the reviewer for this insightful question. To investigate, we continued training Llama for an additional 250, 500, 750, and 1000 steps beyond our original checkpoints:
>
> | Continued Steps | 0     | 250   | 500   | 750   | 1000  |
> | --------------- | ----- | ----- | ----- | ----- | ----- |
> | MATH 500        | 58.04 | 58.32 | 57.96 | 58.58 | 58.47 |
> | GPQA            | 28.28 | 29.21 | 28.72 | 29.03 | 28.62 |
>
> Although Figure 6 suggests that the ratio of solvable problems continues to increase slowly, we observe that additional training yields only slight improvements in downstream performance.
>
> &nbsp;
> &nbsp;
>
> > \[Q7\] Figure 6: it would be interesting to augment this study to test if more direct hints improve the solvable fraction on train data but not test data.
>
> We agree that analyzing whether more direct hints improve the solvable fraction on training versus test data is an interesting direction. For clarity, the solve rate shown in Figure 6 is measured on the training data, not the test data. We have made this explicit in the revised PDF (highlighted in blue).
>
> Below, we report the solve rates on the training data using different types of hints:
>
> | Hint Type     | Solve Rate (%) | MATH 500 Acc |
> | ------------- | -------------- | ----------------- |
> | Abstract Cues | 70             | 58.0              |
> | Partial Steps | 68             | 57.6              |
> | Explanations  | 73             | 54.3              |
> | Ground Truth  | 100            | 27.6              |
>
> We observe only slight differences between Abstract Cues, Partial Steps, and Explanations, and their downstream performance remains relatively similar. In contrast, **using the ground truth as a hint allows the model to “hack” the reward**: the solve rate reaches 100% because the model can directly copy the answer, but the downstream performance drops significantly. This suggests that overly direct hints may harm generalizability.

---

### Author Response · Authors · 2025-11-24
**General Response and Summary of Revisions**

We sincerely appreciate the reviewers’ time and effort in reviewing our manuscript. As highlighted by reviewers, NuRL addresses a timely and important limitation of existing reasoning-oriented RL methods **(RKzS)**. Our paper provides a well-motivated design that allows models to benefit from hints **(GxNS)**. The method is both novel and intuitive **(1SjR)**, with ablations that thoroughly investigate the underlying mechanisms **(1qwq)**. Under a comprehensive experimental setup with carefully designed evaluations **(1qwq, RKzS)**, NuRL consistently improves performance across key reasoning benchmarks **(1qwq, GxNS, RKzS)**, yielding significant gains in both pass@1 and pass@1024 **(1qwq)**. Overall, NuRL offers a simple, well-motivated, and empirically strong approach to advancing RL-based reasoning in LLMs **(1qwq, GxNS, RKzS, 1SjR)**.

We also appreciate **Reviewer 1qwq’s comment** that “*I would be happy to improve my rating after some minor changes that reinforce the paper’s existing strengths*,” as well as **Reviewer 1SjR’s note**: “*If you can make this paper more sound and answer the questions above with solid results, I will raise my score and vouch for you.*”

---

### **Summary of Revisions**

In response to the reviewers’ comments, we have carefully revised and strengthened the manuscript with the following additions:

**Major manuscript updates**

* **Reviewer 1SjR:** Added evaluations on seven more challenging benchmarks (Appendix F).
* **Reviewer 1qwq & RKzS:** Updated Figure 3 with more comprehensive results including Qwen.
* **Reviewer 1qwq:** Added related works suggested by the reviewers (Section 2).
* **Reviewer 1SjR:** Clarified that Figure 6 is measured on the training data.
* **Reviewer RKzS:** Provided a qualitative example comparing GPRO and NuRL outputs (Appendix G).

These manuscript changes are highlighted in *blue* for easy reference.

**Additional experiments and analyses in the response**

* **Reviewer 1qwq:** Added an additional experiment with 1000-step extended training.
* **Reviewer 1qwq:** Added an analysis of solve-rate versus hint type, showing that “worse” hints (e.g., ground truth) can achieve high training solve rates but fail to transfer to test time.
* **Reviewer GxNS:** Added a baseline with dynamic sampling from DAPO, demonstrating that **NuRL’s gains are not solely from training stabilization.**
* **Reviewer GxNS & 1SjR:** Added a baseline comparing offline GRPO, showing that hints capture nearly the same improvement as full reasoning traces while remaining lightweight.
  **Reviewer RKzS & 1SjR:** Added analysis of train–test distribution mismatch, justifying that training with hints and testing without hints is effective.
* **Reviewer 1SjR:** Added **seven more challenging benchmarks across three models**, showing **NuRL consistently outperforms GRPO** (aligning with Table 1).
* **Reviewer 1SjR:** Added qualitative examples showing that NuRL and GRPO have similar output formats and that NuRL does *not* produce hints at test time.

---

We believe these revisions fully address the reviewers' concerns, and we hope they meet the expectations outlined by reviewers 1qwq and 1SjR regarding score updates.

**Authors**

---

### Meta-Review · Area_Chair_ZMhV · 2026-01-09

**Summary:**

The decision is primarily informed by concerns regarding whether NuRL's improvements stem from the hints themselves or merely training stabilization, the potential train-test distribution shift , and the need for evaluation on more rigorous, "elite-level" benchmarks.

**Reviewer Concerns:**

Addressed: Source of Improvement; Harder Benchmarks: Results on additional elite math benchmarks  demonstrated NuRL's sustained advantage; Distribution Shift: Authors clarified that hints are auxiliary signals that model internalizes without developing inference-time dependency.

Outstanding: Scalability: Verification on very large model families is still lacking to confirm impact on stronger base policies.

**Reviewer Scores:**

•	Reviewer 1qwq: 6

•	Reviewer GxNS: 6

•	Reviewer RKzS: 6

•	Reviewer 1SjR: 6

---

### Decision · Program_Chairs · 2026-01-26

Accept (Poster)